# Improving People’s Self-Reported Experience with the Health Services: The Role of Non-Clinical Factors

**DOI:** 10.3390/ijerph17010178

**Published:** 2019-12-25

**Authors:** Ángel Fernández-Pérez, Ángeles Sánchez

**Affiliations:** Department of Applied Economics, University of Granada (Spain), Faculty of Economics and Business Science, Campus Cartuja s/n, 18071 Granada, Spain; sancheza@ugr.es

**Keywords:** public policies, health economics, responsiveness, health system performance, primary care, specialised care, hospital care, POLS

## Abstract

The main aim of this study was to analyse the association between non-clinical factors and the self-reported experience of people with the main health services of the Spanish public healthcare system. Specifically, we analysed whether factors such as the treatment received from health staff, the confidence transmitted to the patient by the doctor, or waiting time for a diagnostic test had an influence on people reporting a more satisfactory experience with primary, specialised, and hospital care services. We used cross-sectional microdata from the Spanish Healthcare Barometer survey of 2015 comprising a sample of 7800 individuals. We applied a probit-adapted ordinary least squares estimation, which is one of the most widely used methods in recent studies on subjective well-being. Our findings suggest that individuals’ interaction with non-clinical factors was positively correlated with the overall health services experience. Treatment received from health staff was one of the most relevant factors to ensure that individuals report a more satisfactory experience with primary care. Time devoted by physicians to each patient and waiting time for a non-emergency admission were the most correlated factors in specialised and hospital care services, respectively. This study could have implications for public policies. First, it shows policy-makers the influence of non-clinical factors when individuals rate their overall experience with the main health services in Spain. Second, it identifies the key factors where the health system could reallocate more public resources to improve people’s experience and thus the health system responsiveness.

## 1. Introduction

In 2000, the World Health Organization (WHO) published Health Systems: Improving Performance, a report that underscored the pressing need to assess health system performance [1]. In the report, the WHO developed a framework for assessing the overall performance of a health system taking into account the following items: level and distribution of health, level and distribution of responsiveness, and fairness in financial contribution. Of these items, those related to responsiveness have received increasing attention in the literature [2,3,4,5]. Health system responsiveness concerns the way individuals are treated and the environment in which they are treated by the health system. Responsiveness is related to the so-called non-clinical factors, an aspect that, although not directly related to health outcomes, may be equally important to guarantee the well-being of the population [6,7]. In addition to ensuring people’s health, treating them with dignity, taking into account their opinion, providing them proper information about their health problems, or attending to them promptly are fundamental objectives of any health system. The WHO has developed eight domains in which most non-clinical factors can be classified: autonomy, choice, communication, confidentiality, dignity, prompt attention, quality of basic amenities, and access to family and social support [6].

The importance of responsiveness stems mainly from its relation to people’s health [6,8,9]. Ensuring better experiences with non-clinical factors could lead individuals to be more cooperative with their health problems, accept treatment procedures, or follow the advice of health staff [10,11]. In developing countries, it has been shown that a lack of responsiveness in health systems can lead to an underutilisation of the health services and cause declining health [12]. Additionally, responsiveness is fundamental due to its relationship with human and patients’ rights [13]. 

As some authors have stated, it is difficult to use objective indicators to measure non-clinical factors [6]. For instance, it is not easy to measure the level of dignity or autonomy with which a health system treats its population in a direct way with an objective indicator. Therefore, the experiences that people report with regard to non-clinical factors have typically been used to quantify the level of health system responsiveness in these domains. Some authors have shown that subjective measures may be a good proxy of the real responsiveness of health systems [2].

Given the importance of responsiveness, all the factors or domains related to this concept should be addressed by health systems. However, recent budgetary constraints in health care provision in the wake of the economic crisis have led to a growing concern about the efficient use of public resources [14,15]. Accordingly, health policy-makers have to decide which non-clinical factors are a priority in order to receive more resources to improve users’ experience with the health system. For instance, should we invest more in reducing waiting times, or would it be better to invest more in improving the communication skills of doctors?

The main aim of this paper was to analyse the association between non-clinical factors and the self-reported experience of people with each of the main health services of the Spanish public healthcare system. Specifically, we identified the most influential non-clinical factors to ensure patients report an overall satisfactory experience with primary (general practitioner), specialised (outpatient care), and hospital care (inpatient care) services. We have also checked whether these factors are the same across health services or if they change depending on the specific service analysed. In addition to measuring the level of responsiveness with each of the non-clinical factors, the findings of this study provide empirical evidence about the role that these factors play in shaping people’s overall experience with the main health services in Spain. In this vein, these results could aid policy-makers in determining which non-clinical factors should be allocated more resources to significantly improve patients’ experience with each health service and hence enhance the overall performance of the Spanish health system. Public health expenditure in Spain accounted for 71.1% of total health expenditure in 2015, whereas private spending accounted for the remaining 28.9% [16]. Therefore, our analysis covered a significant part of the Spanish health system. 

To the best of our knowledge, this is the first study to analyse the role of non-clinical factors for three of Spain’s main health services. Although several studies have examined the influence of non-clinical factors on patient satisfaction, the analyses were confined to certain regional health services [17,18] or to a specific health service [19,20,21,22]. Studies addressing self-reported levels of overall satisfaction with the health system or its services in Spain have primarily examined the influence of socioeconomic or organisational factors. However, the potential effects of non-clinical factors on overall satisfaction have not yet been analysed [23]. Likewise, from the international point of view, studies taking into account non-clinical factors have mainly focused on analysing the influence of these factors on patients’ satisfaction with a health service [24,25,26].

## 2. Materials and Methods

This study uses cross-sectional microdata obtained from the Spanish Healthcare Barometer (SHB) survey of 2015 conducted by the Spanish Centre for Sociological Research in coordination with the Spanish Ministry of Health since 1993. This annual survey collects the opinions of citizens 18 years of age and over on several aspects of the health services of the Spanish National Health System (SNS). Data are gathered using a structured questionnaire administered by means of a personal interview in the respondents’ households. The annual survey is conducted in three waves with a total sample of about 7800 respondents and is representative of the Spanish adult population with a sample error of ±1.14%. The survey uses a multistage sampling procedure stratified by cluster with randomly selected sampling units.

The principle feature of the SHB survey is that the respondents are asked to assess their overall experience with the main Spanish public health services (primary, specialised, and hospital care). The respondents are asked the following question: ‘Based on your own experience or opinion, please rate the following public health services: primary care consultations; specialised care consultations; admission and care in public hospitals’. Respondents rate their experience with each health service on a scale of 1 to 10 where 1 is ‘completely unsatisfactory’ and 10 is ‘completely satisfactory’. 

Likewise, the survey provides information on the degree to which the health services respond to a series of non-clinical factors with which citizens interact. Specifically, we analysed nine items for primary care, seven for specialised care, and six for hospital care These non-clinical factors were assessed according to the respondents’ experience or opinion with each factor on a 10-point scale where 1 is ‘completely unsatisfactory’ and 10 is ‘completely satisfactory’. Table 1 shows the 13 non-clinical factors analysed in this study (Column 1), the questions that respondents answered in the SHB survey to rate their experience with each of the 13 non-clinical factors (Column 2) as well as the health services each non-clinical factor refers to (Column 3).

We also used a set of the respondents’ socioeconomic characteristics commonly applied in the literature and provided by the SHB survey. Specifically, we considered the following 10 variables: age (Aged 60); gender (Female); level of education (Higher education); marital status (Single); place of residence (Urban); place of birth (Born abroad); occupational status (Employed); self-reported health (Good health); self-reported chronic illness (Chronically ill); and experience with public health service (see Table 2).

In order to determine the association between non-clinical factors and the self-reported experience of respondents with each of the three Spanish health services (i.e., primary, specialised, and hospital care), we propose the following model:
(1)Y=α+Xβ+Zρ+Rλ+ε
where Y is the vector of the dependent variable self-reported experience with the health service of n individuals and X is a matrix containing the set of non-clinical factors. Let β denote a vector of parameters where β = (β_1_,…, β_9_)’ is primary care, β = (β_1_,…, β_7_)’ is specialised care, and β = (β_1_,…, β_6_)’ is hospital care. Z denotes the socioeconomic characteristics; ρ denotes a vector of parameters ρ = (ρ_1_,…, ρ_10_)’; R is the set of regional dummies; and ε is the error term that is assumed to have a normal distribution of zero mean and σ^2^ variance. 

Given that we analysed three health services, we ran model (1) three times, one for each health service. We worked with 6252 observations for primary care, 5854 for specialised care, and 4702 for hospital care. We included all respondents without missing values in the studied variables. 

Our parameters of interest were represented by vector β in model (1), which indicates the level of correlation between the self-reported experience with each non-clinical factor and the overall self-reported experience with each health service.

We estimated model (1) using probit-adapted ordinary least squares (POLS) [27]. POLS is a method that is being increasingly used in the most recent literature on subjective well-being [28,29,30,31,32]. The POLS method estimates coefficients using ordinary least squares (OLS) instead of an ordered probit (OP) or ordered logit (OL) method for regression models where the dependent variable is ordered categorically. This method has several outstanding advantages. Whereas POLS and OP yield roughly the same results regarding the sign and significant of the coefficients [27], POLS requires much less computing time and is much easier to understand. First, it facilitates the interpretation of coefficients since they can be directly interpreted as OLS coefficients instead of probabilities. Second, when the model considers the interaction between explanatory variables, its interpretation is not easy in the OP or OL models, whereas POLS facilitates the interpretation [30]. Finally, the cardinalisation process can also be applied in explanatory variables representing ordered categories [27,28,29]. For our study, this aspect is crucial. If the OP or OL method is used instead of POLS, nine dummies (10 categories minus 1) must be included in Equation (1) as explanatory variables for each of the non-clinical factors analysed in the model. For example, in the case of primary care, it would be necessary to include 81 dummy variables (nine non-clinical factors multiplied by nine dummies).

The POLS method is applied in two stages. In the first stage, the dependent and explanatory variables are changed from ordinal to cardinal values, known as the ‘cardinalisation process’. In the second stage, the OLS estimation is applied to the new transformed variables.

In the cardinalisation process, the POLS method draws on the implicit cardinalisation of y_i_* (a continuous unobserved variable) of the latent variable model of the OP or OL methods to transform the observed variable (y), which only takes ordered integer values from 1 to 10 (our response categories), into a variable that is able to take any value on the real line (−∞, +∞) ($\overset{ˉ}{y}$). In order to cardinalise, it is necessary to carry out an increasing monotonic transformation to preserve the order of the response categories of the variables. In line with [27] (pp. 28–34), we used the normal distribution in our study. For this reason, we assumed that our three dependent variables Y will be approximately normally distributed. Figure 1 shows the distribution of frequencies of the overall self-reported experience with each of the health services. The three services were skewed slightly towards positive values although their distributions were similar to the normal distribution. Regarding primary care, more than 90% of the respondents rated the service with a score of 5 or more and around 50% of the respondents gave a rating of 8, 9, and 10. In specialised and hospital care, the values were more centred, since most of the responses (70%) were in the central categories 5–8.

The transformation was performed taking into account the relative and absolute frequencies of the J response categories of the ordered variables as well as the values of the standard normal distribution function. We first obtained the µ_j_ values (cut-points in the OP method). These values were associated to the standard normal distribution function from accumulated frequencies of the J response categories of the ordered categorical variables {µ_j_}^J^_j_=0 with µ_0_ = −∞ and µ_J_ = +∞. Then, we calculated the conditional expectation of the unobserved variable for each of the response categories. Therefore, if the continuous unobserved variable of individual i is y_i_*, where the observed variable is y_i_ = j if µ_j-1_ < y_i_*< µ_j_ for all j = 1, …, J, in accordance with the normal distribution theory, the conditional expectation of the unobserved variable would be:
(2)$${\overset{ˉ}{y}}_{i}=E(y_{i}^{*}|\mu_{j-1}<y_{i}^{*}<\mu_{j})=\frac{n\left(\mu_{j-1}\right)-n\left(\mu_{j}\right)}{N\left(\mu_{j}\right)-N\left(\mu_{j-1}\right)}$$
where ${\overset{ˉ}{y}}_{i}$ is the cardinalisation of the dependent variable; n (·) is the standard normal density function; and N (·) is the accumulated normal distribution function.

Given that non-clinical factors in the X matrix of model (1) are also measured on a scale of 1 to 10, we also applied Equation (2) in order to cardinalise them. 

## 3. Results

Table 3, Table 4 and Table 5 show the descriptive statistics of all the variables analysed in this study. Primary care was the highest rated health service with a mean score of 7.3, while hospital care showed the worst rating with a mean score of 6.5 (see Table 3). With regard to non-clinical factors, confidence and security (primary care), treatment received from health staff (specialised care), and care and attention delivered by nursing staff (hospital care) were the items with which the respondents had the most satisfactory experience. Conversely, items related to waiting times were the most poorly rated out of all health services (see Table 4).

Table 6 shows the results of the estimations with the regression results for each of the three Spanish public health services analysed: primary, specialised, and hospital care. After checking the corresponding tests, we did not detect problems of heteroskedasticity (corrected by the robust standard errors after using weights), multicollinearity (tested using the variance inflation factor), or omitted variables (Ramsey test). The results of the tests are available upon request.

The results in Table 6 indicate that the coefficients of all the non-clinical factors were positive and statistically significant in all health services, with the exception of advice of doctor in specialised and hospital care, and knowledge and follow-up of health problems in primary care, which were not significant. This indicates that the more satisfactory the self-reported experience with non-clinical factors, the more satisfactory the experience with the health services as a whole.

More specifically, when focusing on the quantitative importance of the correlations (absolute value of the coefficients), not all of the correlations were found to have the same effect on self-reported experience with the health service. In primary care, for instance, we found that both treatment received from health staff (treatment received) and confidence and security transmitted by the doctor (confidence and security) showed the highest correlation, thus indicating that these factors contributed most to people reporting a more satisfactory experience with that health service. The Wald test of equality of the estimated parameters revealed that both coefficients were statistically equal (F [1, 6214] = 1.30; *p* = 0.3525), but different from the third one (F [1, 6214] = 4.93; *p* = 0.0264). From a statistical point of view, four non-clinical factors in specialised care had a similar quantitative importance on self-reported experience with the service as a whole, namely time devoted by the doctor, treatment received, confidence and security, and waiting time for diagnostic tests (Wald joint test of equality of estimated parameters: F [2, 5818] = 0.89, *p* = 0.4087). Finally, in hospital care, three non-clinical factors, waiting time for non-emergency admission, information received on health problems, and care and attention by medical staff showed the highest association for self-reported experience with the service in similar quantitative terms (Wald joint test of equality of estimated parameters: F [2, 4667] = 0.30, *p* = 0.7404).

Regarding the socioeconomic variables, for the sake of simplicity, in Table 6 we omitted the coefficients of the socioeconomic variables, which were not significant for any of the three health services. This was the case of age, marital status, employment status, chronic illness, and experience. In line with the literature [33,34], our findings indicated that some individuals’ socioeconomic characteristics were correlated with self-reported experience of the health service. For example, gender was statistically significant only for primary care. Additionally, females reported a more satisfactory experience with primary care than males, which may be due to the utilisation patterns of the services [35]. Regarding educational level, people with a higher education reported a worse overall experience with primary and hospital care services, perhaps because of their higher expectations [36]. Respondents living in urban areas (towns with more than 10,000 inhabitants) tended to report a better overall experience in primary care. People born outside of Spain assessed their overall experience with the specialised and hospital care services better than those born in Spain, which may be explained by the ‘happy migrant effect’, that is, people from other countries tend to minimise the negative effect of their care and are usually more satisfied with the care they receive than nationals [37]. Finally, people with self-reported good health indicated a better experience with primary care.

## 4. Conclusions

Our findings showed that individuals’ interaction with non-clinical factors was significantly associated with a satisfactory self-reported experience with the main Spanish health services. Therefore, health policy-makers should not only focus on the medical or technical aspects of healthcare, but also on non-clinical factors to ensure that people have a more satisfactory experience with the health services. This is a key finding because, as the literature has emphasised, improving responsiveness has positive effects on people’s health.

The results of our study indicate that not all non-clinical factors correlate in the same manner with the self-reported experience with the health services. For each health service, we identified where the scarce public resources could be targeted to ensure people have a more satisfactory experience with the analysed health service. More specifically, based on our findings in primary care, it is important to improve the treatment and confidence and security transmitted by the health staff. In specialised care, our results suggest that it is necessary to increase the time doctors devote to each patient, enhance the treatment received, improve the confidence and security transmitted by the health staff, and reduce the waiting times for diagnostic tests. It is worth highlighting that there is more room for improvement in waiting times for diagnostic tests and time devoted by doctors, since they are the items with which people are least satisfied, as indicated by the scores for these two factors (5 and 6.7, respectively). 

Finally, in hospital care, it would be convenient to reduce the waiting times for non-emergency admission, provide patients with better information about their health problems, and improve the care and attention delivered by medical staff. If we look more closely at the results for hospital care, despite the importance of waiting times for non-emergency admissions as a driver of a more satisfactory self-reported experience, our findings showed that this service received the lowest rating for this non-clinical factor, with an average score of 4.8 out of 10. This low score could explain, at least partially, why hospital care in Spain is the most poorly rated health service by people (an average score of 6.5).

Likewise, these results indicate that in the Spanish case, the key factors of experience are related to respect for people, which is not so dependent on economic resources [38]. For instance, ensuring respectful treatment by the health staff only requires developing certain personal skills that could be fostered through better training in higher education programmes. Conversely, improving waiting times or increasing the time devoted to each patient could be more resource demanding since more health staff or amenities would be needed. To sum up, policy-makers should consider the type of financing, the way in which the resources are combined, and the development of strategies for generating resources that could contribute to ensuring a better attainment of crucial non-clinical factors [39].

Regarding the limitations of our study, it is worth noting that, due to the lack of data, we have not been able to include other non-clinical factors that could be associated with self-reported experience, nor have we been able to analyse the same factors in all health services. Furthermore, due to cultural idiosyncrasies as well as differences in the conception of the welfare state across populations, it would not be appropriate to extrapolate these results to other countries. For this reason, further studies that examine the influence of non-clinical factors on performance in different settings as well as from a dynamic perspective over time are needed.

## Figures and Tables

**Figure 1 ijerph-17-00178-f001:**
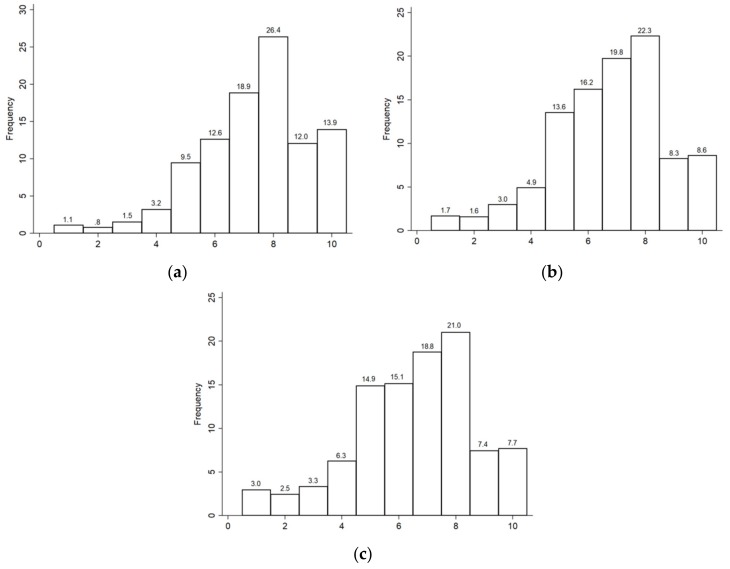
Distribution of frequencies of the overall self-reported experience with the Spanish health services: (**a**) Primary care; (**b**) Specialised care; (**c**) Hospital care. Data from the Spanish Centre for Sociological Research, Spanish Healthcare Barometer survey, 2015.

**Table 1 ijerph-17-00178-t001:** Non-clinical factors included in the Spanish Healthcare Barometer by health service in 2015.

Non-Clinical Factor	Question in the SHB Survey: Based on Your Own Experience or Opinion, Please Rate the Following Items: …	Health Service ^1^
Treatment received	The treatment received from the health staff	Primary Specialised
Time devoted by doctor	The time devoted by the doctor to each patient	Primary Specialised
Confidence and security	The confidence and security transmitted by the doctor	Primary Specialised
Knowledge and follow-up of health problems	Knowledge of medical records and follow-up of health problems	Primary
Information received on health problem	The information received on your health problem	Primary Specialised Hospital
Advice of doctor	Advice of the doctor about exercise, diet, smoking, alcohol consumption, etc.	Primary Specialised Hospital
Waiting time for appointments	The time you have to wait since you make the appointment until you are seen by the doctor	Primary Specialised
Waiting time for diagnostic tests	The waiting time for carrying out diagnostic tests	Primary Specialised
Care by nursing staff	The care delivered by nursing staff	Primary
Waiting time for non-emergency admission	The waiting time for a non-emergency admission	Hospital
Care and attention by medical staff	The care and attention delivered by medical staff	Hospital
Care and attention by nursing staff	The care and attention delivered by nursing staff	Hospital
Number of people sharing room	The number of people who share a room	Hospital

Note: SHB = Spanish Healthcare Barometer. Adapted from the Spanish Centre for Sociological Research. ^1^ Respondents are asked to respond to the questions in reference to the three health services.

**Table 2 ijerph-17-00178-t002:** Socioeconomic variables of respondents to analyse self-reported experience with the public Spanish health services in 2015.

Variable	Definition
Aged60	1 if respondent is aged over 60 and 0 otherwise
Female	1 = female, 0 = male
Higher education	1 if respondent has secondary or tertiary education and 0 if respondent has primary education or no schooling
Single	1 if respondent is single and 0 if respondent is married, widowed, separated, or divorced
Urban	1 if respondent lives in a city with over 10,000 inhabitants and 0 otherwise (rural)
Born abroad	1 if respondent was not born in Spain and 0 otherwise
Employed	1 if respondent is employed and 0 otherwise
Good health	1 if respondent perceives his/her state of health as good or very good and 0 otherwise
Chronically ill	1 if respondent reports being chronically ill and 0 otherwise
Experience with public health service	1 if respondent has used, at least once, the Spanish public health service in the last 12 months and 0 otherwise

Note: Centre for Sociological Research, Spanish Healthcare Barometer survey, 2015.

**Table 3 ijerph-17-00178-t003:** Descriptive statistics of the respondents’ self-reported experience with each Spanish health service in 2015.

Type of Healthcare	N	Mean	SD	Min–Max
Primary	6252	7.3	1.8	1–10
Specialised	5854	6.8	2.0	1–10
Hospital	4702	6.5	2.1	1–10

Note: SD = Standard deviation; Min = Minimum; Max = Maximum. Adapted from the Spanish Centre for Sociological Research, Spanish Healthcare Barometer survey, 2015.

**Table 4 ijerph-17-00178-t004:** Descriptive statistics of non-clinical factors by public health service in Spain in 2015.

Non-Clinical Factors	Mean	SD
*Primary care* ^1^		
Treatment received	7.6	1.7
Time devoted by doctor	7.1	2.0
Confidence and security	7.7	1.9
Knowledge and follow-up of health problems	7.5	1.9
Information received on health problem	7.5	1.9
Advice of doctor	7.4	2.1
Waiting time for appointments	6.6	2.2
Waiting time for diagnostic tests	5.7	2.3
Care by nursing staff	7.4	1.8
*Specialised care* ^2^		
Treatment received	7.3	1.8
Time devoted by doctor	6.7	1.9
Confidence and security	7.2	1.9
Information received on health problem	7.1	2.0
Advice of doctor	6.9	2.1
Waiting time for appointments	5.1	2.3
Waiting time for diagnostic tests	5.0	2.3
*Hospital care* ^3^		
Waiting time for non-emergency admission	4.8	2.2
Care and attention by medical staff	7.2	1.9
Care and attention by nursing staff	7.3	1.8
Number of people sharing room	5.7	2.3
Information received on health problem	7.2	1.9
Advice of doctor	7.0	2.0

Note: SD = Standard deviation. Responses are rated on a scale of 1–10 where 1 is ‘completely unsatisfactory’ and 10 is ‘completely satisfactory’. Adapted from the Spanish Centre for Sociological Research, Spanish Healthcare Barometer survey, 2015. ^1^ N = 6252; ^2^ N = 5854; ^3^ N = 4702.

**Table 5 ijerph-17-00178-t005:** Descriptive statistics of respondents’ socioeconomic characteristics, 2015.

Variables	Health Service					
	Primary ^1^		Specialised ^2^		Hospital ^3^	
	Mean	SD	Mean	SD	Mean	SD
Aged60	0.28	0.45	0.28	0.45	0.28	0.45
Female	0.52	0.50	0.53	0.50	0.52	0.50
Higher education	0.75	0.43	0.75	0.43	0.76	0.43
Single	0.31	0.46	0.31	0.46	0.30	0.46
Urban	0.79	0.40	0.80	0.40	0.79	0.40
Born abroad	0.11	0.31	0.10	0.30	0.10	0.30
Employed	0.43	0.49	0.43	0.50	0.44	0.50
Good health	0.74	0.44	0.73	0.45	0.73	0.44
Chronically ill	0.32	0.47	0.33	0.47	0.33	0.47
Experience with public						
Primary care	0.72	0.45				
Specialised care			0.42	0.49		
Hospital care					0.09	0.29

Note: SD = Standard deviation. Adapted from the Spanish Centre for Sociological Research, Spanish Healthcare Barometer survey, 2015. ^1^ N = 6252; ^2^ N = 5854; ^3^ N = 4702.

**Table 6 ijerph-17-00178-t006:** Determinants of overall self-reported experience with the Spanish public health services in 2015.

	Health Service		
	Primary	Specialised	Hospital
*Non-clinical factors*			
Advice of doctor	0.054 *	0.031	0.042
	(0.022)	(0.023)	(0.025)
Confidence and security	0.172 ***	0.121 ***	
	(0.027)	(0.030)	
Time devoted by doctor	0.095 ***	0.164 ***	
	(0.021)	(0.024)	
Knowledge and follow-up of health problems	−0.012		
	(0.026)		
Information received on health problem	0.128 ***	0.085 **	0.175 ***
	(0.029)	(0.029)	(0.028)
Treatment received	0.215 ***	0.127 ***	
	(0.021)	(0.026)	
Care by nursing staff	0.063 ***		
	(0.019)		
Care and attention by medical staff			0.152 ***
			(0.030)
Care and attention by nursing staff			0.124 ***
			(0.028)
Waiting time for appointments	0.084 ***	0.103 ***	
	(0.018)	(0.025)	
Waiting time for diagnostic tests	0.039 *	0.116 ***	
	(0.016)	(0.025)	
Waiting time for non-emergency admission			0.181 ***
			(0.017)
Number of people sharing room			0.001 ***
			(0.017)
*Socioeconomic variables*			
Female	0.039 *	−0.021	−0.011
	(0.019)	(0.022)	(0.025)
Higher education	−0.093 ***	−0.058	−0.081 *
	(0.028)	(0.033)	(0.040)
Urban	0.074 **	0.035	0.016
	(0.025)	(0.028)	(0.030)
Born abroad	0.026	0.134 ***	0.097 *
	(0.033)	(0.039)	(0.046)
Good health	0.056 *	0.020	0.036
	(0.026)	(0.030)	(0.034)
Aged 60	Yes	Yes	Yes
Single	Yes	Yes	Yes
Employed	Yes	Yes	Yes
Chronically ill	Yes	Yes	Yes
Experience with public health services	Yes	Yes	Yes
*Regional variables* ^1^	Yes	Yes	Yes
Constant	−0.121 *	−0.112	−0.184 **
	(0.053)	(0.058)	(0.066)
Observations	6252	5854	4702
R^2^	0.537	0.422	0.431

Note: Probit-adapted ordinary least squares regression. Entries show parameter estimates with robust standard errors in parenthesis. ‘Yes’ indicates that the variables have been included in the model, but their coefficients were not statistically significant in any of the regressions. ^1^ Some coefficients of the region dummies (16 dummies) were significant but have not been included for the sake of brevity. *** *p* < 0.001; ** *p* < 0.01; * *p* < 0.05.

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
