# Peer review of "Improving People’s Self-Reported Experience with the Health Services: The Role of Non-Clinical Factors"

_ijerph, 2019, doi:10.3390/ijerph17010178_

Round 1
Reviewer 1 Report
Corrections to minor methodological errors are described in the attached .pdf file.

Reviewer 2 Report
This is a well-written manuscript about an interesting study. I have a few major comments and several comments on details.
Main comments
The participants answered questions about various aspects of a medical system (primary care, hospital care and specialized care) and a general question about their satisfaction with the care system. Though the authors mention which aspects were questioned, the actual questions were not mentioned. They could be presented in an additional file or in the main text. The authors based their clinical advices on the highest correlation between the scores to specific aspects and the general question. This is understandable as these aspects have apparently the highest impact on satisfaction. However, if participants are mostly satisfied with certain aspects, it is useless to advice more attention for that aspect, even if its correlation with general satisfaction is very high. On the other hand, of participants are not very satisfied with a certain aspect, it may be advisable to give that aspect more attention, even if the correlation is not the highest.
Minor comments
Introduction, l. 50-51, “Due to the intrinsic features of non-clinical factors, it is difficult to use objective indicators to measure these factors”: I would suggest to delete the part of the sentence before the comma. In my opinion “the intrinsic features” is wrongly used here. Materials and methods, l. 84, “cross-section microdata”: What are microdata? Should it not be cross-sectional”? L. 87-88: There is not much information about the way data was collected. Which measures were taken by the interviewers to avoid socially desirable answers? Where interviewees asked to mention concrete examples of their interactions with the health care systems? L. 116-118: “Given that our dependent variable is a subjective measure, we control for a set of the respondents’ socioeconomic characteristics commonly used in the literature”: If the dependent variable was objective, there could also be reasons to control for confounding factors. They authors used probit-adapted ordinary least squares. Their – not very convincing - argument for choosing this method is that it was used in several other publications in this area. However, they should refer to a publication that has statistically shown that this is the best method for their analysis. L. 126-127: “it facilitates the interpretation of coefficients since they can be directly interpreted as OLS coefficients”: This is not a strong argument for preferring POLS above OLS. I cannot read the right part of the table on Page 4. L. 166: On a scale from 1 to 10, it is exaggerated to use to decimals for a score (7.31, for instance).
Reviewer 3 Report
It is an interesting and meaningful paper to analyze the public’s perception on different non-clinical factors which affect their satisfaction of health services. Several points need clarification and elaboration.
In the Abstract, it is unclear what doctor-patient confidence means.There may be some overlapping in definitions between specialized care and hospital care. Does specialized care here mean outpatient specialized services?
As waiting time was poorly rated, it would be helpful to describe in the Discussion the usual waiting time for primary, specialized care and non-emergency hospital services in Spain. How is the waiting time compared with other countries? This will help international readers to have a better understanding of the context in Spain.
It is also helpful to describe in the Introduction the ratio of public health services to private services in Spain.
The Conclusion section should be renamed as Discussion.
Reviewer 4 Report
Rows 1-25-Check tenses.
Rows 65-67- Check tenses
Table 1- Much of this Table did not fit on the page. It should have been placed earlier in the text.
Table 6 reads as confusing. This needs some clarification.
There are many writing and wording errors that include, tenses, spelling, and clarity.
